

# Development of molecular identification methods for *Dryophytes suweonensis* and *D. japonicus*, and their hybrids

Nakyung Yoo[1], Ju-Duk Yoon[1], Jeongwoo Yoo[1], Keun-Yong Kim[2], Jung Soo Heo[2] and Keun-Sik Kim[1]

[1] Research Center for Endangered Species, National Institute of Ecology, Yeongyang, Republic of Korea
[2] Department of Genetic Analysis, AquaGenTech Co., Ltd, Busan, Republic of Korea

## ABSTRACT

**Background.** As hybridization can reduce biodiversity or cause extinction, it is important to identify both purebred parental species and their hybrids prior to conserving them. The Suwon tree frog, *Dryophytes suweonensis*, is an endangered wildlife species in Korea that shares its habitat and often hybridizes with the Japanese tree frog, *D. japonicus*. In particular, *D. suweonensis*, *D. japonicus*, and their hybrids often have abnormal ovaries and gonads, which are known causes that could threaten their existence.

**Methods.** We collected 57 individuals from six localities where *D. suweonensis* is known to be present. High-resolution melting curve (HRM) analysis of the mitochondrial 12S ribosomal RNA gene was performed to determine the maternal species. Thereafter, the DNA sequences of five nuclear genes (*SIAH*, *TYR*, *POMC*, *RAG1*, and *C-MYC*) were analyzed to determine their parental species and hybrid status.

**Results.** The HRM analysis showed that the melting temperature of *D. suweonensis* was in the range of 79.0–79.3 °C, and that of *D. japonicus* was 77.7–78.0 °C, which clearly distinguished the two tree frog species. DNA sequencing of the five nuclear genes revealed 37 single-nucleotide polymorphism (SNP) sites, and STRUCTURE analysis showed a two-group structure as the most likely grouping solution. No heterozygous position in the purebred parental sequences with Q values ≥ 0.995 were found, which clearly distinguished the two treefrog species from their hybrids; 11 individuals were found to be *D. suweonensis*, eight were found to be *D. japonicus*, and the remaining 38 individuals were found to be hybrids.

**Conclusion.** Thus, it was possible to unambiguously identify the parental species and their hybrids using HRM analysis and DNA sequencing methods. This study provided fundamental information for *D. suweonensis* conservation and restoration research.

Corresponding authors
Ju-Duk Yoon, grandblue@nie.re.kr
Keun-Sik Kim, kskim@nie.re.kr

## INTRODUCTION

Hybridization is the reproduction between two genetically distinct species (*Barton & Hewitt, 1985*). It is caused due to human activities such as the introduction of plant or animal species or habitat fragmentation and modification (*Grabenstein & Taylor,*

*2018*). The more rapidly these activities interact, the more rapidly hybridization occurs (*Rhymer & Simberloff, 1996*). This can cause outbreeding depression, which in severe cases can lead to species extinction and reduced biodiversity (*Hoffmann et al., 2015*; *Huff et al., 2011*). Moreover, hybrids may be less stable than purebreds because of interspecific incompatibities or various negative effects (*Coyne & Orr, 2004*; *Moulia, 1999*). Hybrid individuals that inherit half the genes from each parental species are often morphologically indistinguishable from their parents (*Leary, Gould & Sage, 1996*). It is currently estimated that hybridization occurs in approximately 10% of animal species (Lepidoptera: Rhopalocera and Heliconiina, Paradisaeidae and Paridae, *etc.*), although the actual percentage is likely to be higher because most hybrids are difficult to identify in the wild (*Mallet, 2005*). Moreover, hybridization is common in frogs (*Berger, 1968*; *Kierzkowski et al., 2013*; *Peek et al., 2019*). Identifying hybrids is important as specific species population can be restored by removing hybrid individuals or by captive breeding if a population contains a sufficient number of parental individuals without hybrids (*Allendorf et al., 2001*).

The Suwon tree frog, *Dryophytes suweonensis*, has become an endangered wildlife species because of population fragmentation, hybridization, competition, and continued habitat loss (*Borzée, 2018*; *Zhang et al., 2019*). As a result, this species is designated as a Class I endangered wildlife in Korea and is listed as Endangered (EN) on the IUCN Red List (*IUCN, 2017*). While the Japanese tree frog, *D. japonicus* uses a variety of habitats, including forests, wetlands, and rice fields, and is widely distributed in Asia, *D. suweonensis* is mainly found in lowland rice field wetlands and is known to be endemic to the Korean Peninsula (*Do et al., 2017*). *D. suweonensis* diverged from *D. japonicus* between 6.4 mya and 5.1 mya and is characterized by very low genetic diversity compared to *D. japonicus* (*Chun et al., 2012*; *Li et al., 2015*).

Purebred species and their hybrids have been identified using a various analytical methods, such as mitochondrial DNA (mtDNA) sequencing, microsatellite analysis, single-nucleotide polymorphism (SNP) analysis, and restriction-site associated DNA capture (rapture) sequencing (*Iwaoka et al., 2021*; *Simoes, Lima & Farias, 2012*; *Melville et al., 2017*; *Peek et al., 2019*). A previous study reported that hybridization has also occurred between *D. suweonensis* and *D. japonicus* in their wild populations by analyzing both mitochondrial cytochrome c oxidase I (COI) and microsatellite markers (*Borzée et al., 2020*). mtDNA is widely used in population genetics to measure genetic variation in various wildlife species to assess population differentiation and habitat conservation strategies (*Avise et al., 1987*; *Moritz, 1994*). However, there are limitations with respect to determining hybridization using mtDNA alone as it only provides information on maternal inheritance (*Sato & Sato, 2013*). Moreover, the use of microsatellite markers from different species can cause errors because of the high probability of null allele occurrences as the taxonomic distance between species increases (*Wan et al., 2004*).

In this study, the high-resolution melting curve (HRM) technique was employed to identify the two tree frog species, *D. suweonensis* and *D. japonicus*, and their hybrids, based on the mitochondrial 12S ribosomal RNA (rRNA) gene, which allowed us to identify their maternal parents (*Yoo et al., 2022*). Moreover, primer sets for five nuclear genes, namely

E3 ubiquitin protein ligase 1 (*SIAH*), tyrosinase (*TYR*), proopiomelanocortin (*POMC*), V(D)J recombination-activating protein 1 (*RAG1*), and transcriptional regulator Myc-like (*C-MYC*) were designed, and single-nucleotide polymorphism (SNP) sites were detected by sequencing their amplicons to determine their parentage and hybridization. This integrated approach facilitated the unambiguous identification of purebred parental species and their hybrids, thus providing valuable information for conservation and restoration research of *D. suweonensis*.

## MATERIALS & METHODS

### Sampling and DNA extraction

From April to June 2021, we sampled a total of 57 tree frog individuals from six localities in South Korea, including Suwon (two) and Pyeongtaek (14) cities in Gyeonggi-do, Chungju City (nine) in Chungcheongbuk-do, Asan City (14) in Chungcheongnam-do, Iksan City (eight) and Wanju County (10) in Jeollabuk-do, where *D. suweonensis* is known to be present (Fig. 1). Surveys were conducted during the day when the tree frogs were found to be active, and they were captured randomly while walking around rice field banks in the vicinity of rice field wetlands, which are the main habitats of this species (*Kim et al., 2012*). To perform molecular experiments, oral epithelial cells were called non-invasively obtained according to *Goldberg, Kaplan & Schwalbe (2003)*, that is, a sterile cotton swab (Han Chang Medic, Cheonan, Korea) was used to gently swab the inside the frog's mouth for approximately 30 s to 1 min. Genomic DNA (gDNA) was extracted using the DNeasy Blood & Tissue Kit (Qiagen, Hilden, Germany) according to the manufacturer's instructions. The amount of extracted gDNA was determined using a spectrophotometer (DeNovix DS-11 FX, DeNovix Inc., Wilmington, USA).

### HRM analysis

HRM analysis of the mitochondrial 12S rRNA gene was performed as previously described by *Yoo et al. (2022)*. Briefly, a total volume of 20 µl PCR reaction was prepared, containing 10 µl of MeltDoctor™ HRM Master Mix (Thermo Fisher Scientific, Waltham, MA, USA), gDNA (10 ng/ µl), and 2 µl of a primer set at 5 µM (HYL-12S-0250f: 5′-GTTACACCACGAGGCTCA-3′ HYL-12S-0343r: 5 ′-TGAGTTTCTTAAGAACAAGCG-3′), with 6 µl of sterile distilled water. The PCR reaction was performed using the QuantStudio 5 Real-Time PCR System (Thermo Fisher Scientific, Waltham, MA, USA) with an initial denaturation step at 95 °C for 10 min, followed by 40 cycles of 95 °C for 15 s and 60 °C for 1 min for annealing/extension. The meltcurve and dissociation steps for HRM analysis were conducted at 95 °C for 10 s for denaturation and 60 °C for 1 min for binding. Subsequently, high-resolution melting was performed at 95 °C for 15 s, followed by 60 °C for 15 s for binding.

For an individual that did not show a reliable melting temperature, its gDNA was PCR amplified using the forward primer 5′-AAAGCRTAGCACTGAAAATG-3′ (ANU-MT-00018f) and the reverse primer 5′-TCGGTGTAAGCGAGATGCTTT-3′ (ANU-MT-01017r). The amplified PCR products were then sequenced using the method

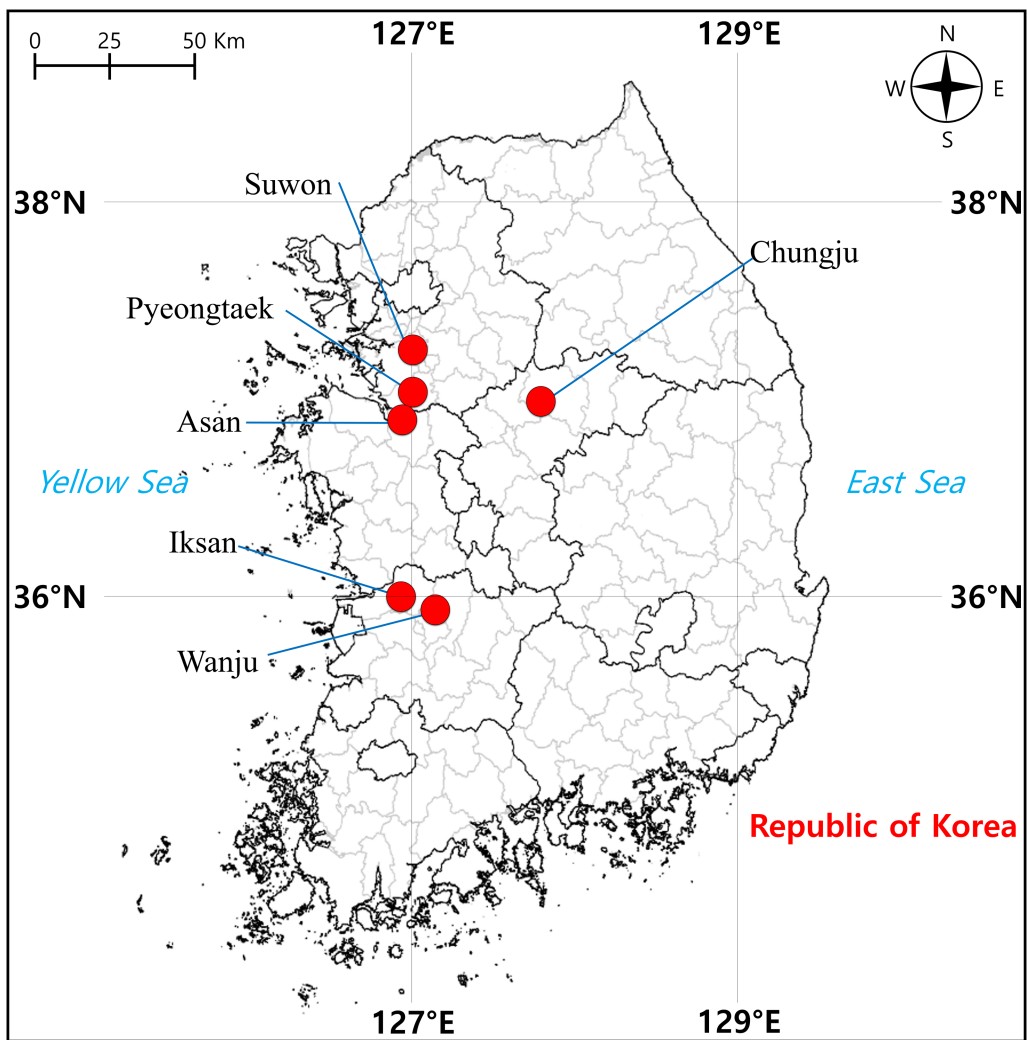

**Figure 1** Sampling localities of individuals of *Dryophytes suweonensis*, *D. japonicus*, and their hybrids in South Korea.

described above and identification was performed using BLASTn in National Center for Biotechnology Information (NCBI, https://www.ncbi.nlm.nih.gov/).

## PCR primer design and DNA sequencing

To design primer sets for PCR amplification of the five nuclear genes, *SIAH*, *TYR*, *POMC*, *RAG1*, and *C-MYC*, the nucleotide sequence information of *Dryophytes* and *Hyla spp.* available in the GenBank database of the NCBI was downloaded.

The nucleotide sequence information of the five nuclear genes was subjected to multiple sequence alignment using ClustalW (*Thompson, Gibson & Higgins, 2003*) in BioEdit 7.2 (https://thalljiscience.github.io/), and five new primer sets were designed based on the information around the highly conserved regions (Table 1). To validate the primer sets, the PCR reactions were carried out with 10 µl of Platinum Hot Start PCR Master Mix 2X

**Table 1** Primer sets newly designed in this study to identify purebred *Dryophytes suweonensis, D. japonicus,* and their hybrids.

| Genes | Primer name | Sequence (5′–>3′)* | Tm (°C) | Total bases (bp) |
|---|---|---|---|---|
| E3 ubiquitin protein ligase 1 (*SIAH*) | DS_SIAH_F | TGGCAAGAAAAACAATATCCTCTC | 60.1 | 24 |
| | DS_SIAH_R | ATGTCAGAGCGGACATCTTGT | 59.4 | 21 |
| Tyrosinase (*TYR*) | DS_TYR_F | TGTGCCAGGGCGCGAAG | 59.6 | 17 |
| | DS_TYR_R | TTAGTGGGATTGACGATMGRGAAA | 60.1 | 24 |
| V(D)J recombination-activating protein 1 (*RAG1*) | DS_RAG1_F | AACCTGTGTGTTTAATGCTGGC | 60.3 | 22 |
| | DS_RAG1_R | TTCGGGCAAAGTTTCCATTCA | 61.1 | 21 |
| Proopiomelanocortin (*POMC*) | DS_POMC_F | AACGTCCGRAAGTACGTCATGA | 60.3 | 22 |
| | DS_POMC_R | CCATCGRAAGTGATGCATTTTGTA | 60.1 | 24 |
| Transcriptional regulator Myc-like (*C-MYC*) | DS_C-MYC_F | TCCAGCCTTTTTCCATCTACTGA | 60.3 | 23 |
| | DS_C-MYC_R | GCTGGTCCTACTGGTTCCTA | 60.5 | 20 |

**Notes.**

*$M = A + C, R = A + G.$

(Invitrogen, Waltham, USA), 100 ng of gDNA, 1 µl of each primer at 5 µM, and the final volume was adjusted to 20 µl using sterilized tertiary distilled water. The PCR reaction consisted of an initial denaturation at 94 °C for 2 min, followed by 38 cycles of denaturation at 94 °C for 30 s, annealing at 56 °C for 30 s, and extension at 72 °C for 30 s. Finally, after an extension step at 72 °C for 1 min, the results obtained were confirmed by electrophoresis on a 2% agarose gel stained with GelRed (Invitrogen, Waltham, MA, USA).

The amplified PCR products were purified using the *AccuPrep*® PCR Purification Kit (Bioneer, Daejeon, Korea) following the user manual. For DNA sequencing, the BigDye™ Terminator v3.1 Cycle Sequencing Kit (Thermo Fisher Scientific, Waltham, MA, USA) and the DNA Analyzer 3730*xl* (Thermo Fisher Scientific, Waltham, MA, USA) were utilized. The forward and reverse primers used in PCR for each nuclear gene were further used for cycle sequencing. Subsequently, the raw data for each nuclear gene were aligned using SEQUENCHER version 5.4.6 (*Nishimura, 2000*), and unnecessary parts were trimmed to complete the contigs.

## STRUCTURE analysis

To identify patterns in the degree of hybridization between *D. suweonensis* and *D. japonicus*, we conducted a STRUCTURE analysis using the Bayesian clustering algorithm. For this analysis, a nucleotide sequence matrix that included both SNPs representing interspecific differences between the two tree frog species and SNPs identifying individual variations were created. SNPs were analyzed using STRUCTURE *v.* 2.3.4. (*Pritchard, Stephens & Donnelly, 2000*) with 100,000 burn-ins and 500,000 simulations. Moreover, posterior probabilities (LnP(D)) values were calculated using the delta K (∆K) method through STRUCTURE HARVESTER (*Evanno, Regnaut & Goudet, 2005*) to determine the optimal K value (*Earl & VonHoldt, 2012*).

**Table 2  Single-nucleotide polymorphism (SNP) sites (left) and parsimony informative sites (PI) (right), separated by a forward slash, in five nuclear genes of purebred *Dryophytes suweonensis*, *D. japonicus*, and their hybrids.**

|  | *SIAH* | *TYR* | *POMC* | *RAG1* | *C-MYC* |
|---|---|---|---|---|---|
| *Dryophytes suweonensis* | 0/0 | 0/0 | 2/0 | 2/0 | 1/1 |
| *Dryophytes japonicus* | 0/0 | 5/4 | 1/1 | 1/0 | 0/0 |
| Hybrids | 3/3 | 4/4 | 7/6 | 9/7 | 3/2 |
| Total | 3/3 | 9/9 | 9/7 | 11/9 | 4/2 |

## RESULTS

### HRM analysis

The HRM analysis of the mitochondrial 12S rRNA gene revealed distinct melting temperatures of 79.0–79.3 °C for *D. suweonensis* and 77.7–78.0 °C for *D. japonicus*, enabling reliable species identification. However, one individual (SJ02_5) had a melting temperature of 78.6 °C, which made it challenging to identify that particular species accurately. The species identification success rate using HRM analysis was approximately 97.88%. DNA sequencing identified SJ02_5 as *D. japonicus* with 99.78% identity to the nucleotide information in the GenBank database (GenBank accession number: OK156173).

### DNA sequencing

DNA sequencing of the five nuclear genes sampled from the 57 individuals of *D. suweonensis* and *D. japonicus* revealed the following specific sequence lengths for each gene; 267 bp for *SIAH*, 361 bp for *TYR*, 372 bp for *POMC*, 561 bp for *RAG1*, and 301 bp for *C-MYC*. When comparing the variable sites and parsimony-informative sites (PIs) of each gene in the two tree frog species and their hybrids, no variable sites and PIs were identified in *TYR* and *SIAH* for *D. suweonensis*, and no PIs were identified in *C-MYC* and *SIAH* for *D. japonicus*. The nuclear genes with the most variable sites overall were *RAG1*, whereas those with the most PIs were *TYR* and *RAG1* (Table 2).

DNA sequencing of the five nuclear genes revealed that their sequence chromatograms displayed heterozygous sequences at the SNPs between the two tree frog species in numerous individuals (Fig. 2). For instance, the hybrid individuals showed a heterozygous sequence of (G/A) at 175 bp in *SIAH*, of (T/C) at 202 bp in *TYR*, of (G/C) and (G/A) at 91 bp and 93 bp in *POMC*, and (A/T) at 97 bp in *RAG1*.

### STRUCTURE analysis

STRUCTURE analysis was performed to calculate the optimal number of groups based on the Q values calculated repeatedly using the STRUCTURE HARVESTER, and the highest delta K value was found at $K = 2$ (Fig. 3A). Using the optimal number of groups $K = 2$, the graph with the lowest maximum likelihood value ($K = 2$, Est. Ln prob. of data = 1,232.8) was selected (Fig. 3B).

In STRUCTURE analysis, the Q value is the estimated probability that each individual belongs to a specific species or population. When the Q value, which is the criterion that no heterozygous sequences appear at the heterozygous mutation positions and is a sequence

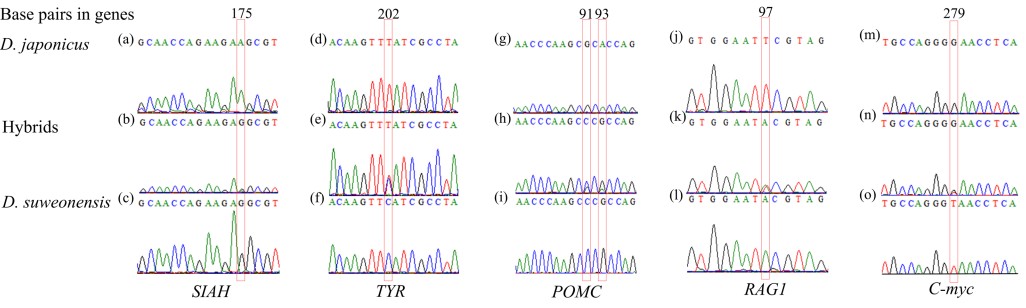

**Figure 2 Examples of heterozygous sequences at single-nucleotide polymorphism (SNP) sites in sequencing chromatograms of five nuclear genes of *Dryophytes suweonensis*, *D. japonicus*, and their hybrids.** (A–C) *SIAH*, (D–F) *TYR*, (G–I) *POMC*, (J–L) *RAG1*, and (M–O) *C-myc* of *D. suweonensis*, *D. japonica*, and their hybrids. Red box represents heterozygous sequences.

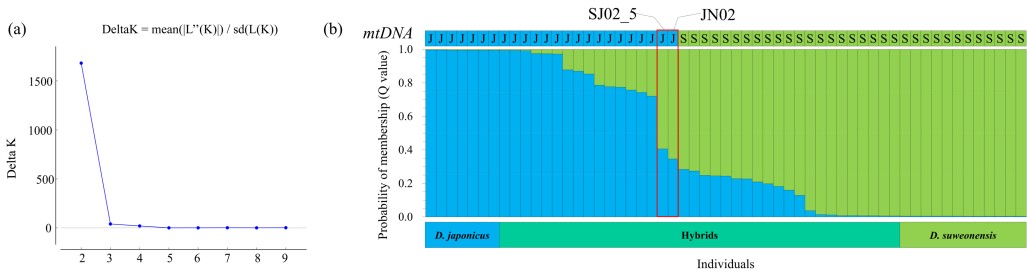

**Figure 3 Identification results of purebred *Dryophytes suweonensis*, *D. japonicus*, and their hybrids.** (A) The number K of groups for *D. suweonensis* and *D. japonicus* obtained by the delta K (ΔK) method in STRUCTURE Harvester. The value of ΔK was highest at 2 (ΔK: 1681.844). (B) Results of HRM analysis of mitochondrial 12S rRNA gene (A captial "S" means *D. suweonensis* and a capital "J" means *D. japonicus*) and probabilistic assignment to genetic clusters (K = 2) using the STRUCTURE software. A vertical column represents each individual, and the length of each column indicates the proportional membership (Q value) in each cluster (*D. suweonensis* is green, *D. japonicus* is blue, and red box represents highly admixed individuals).

that shows SNPs between species among the five nuclear gene sequences, was determined to be 0.995 or higher, 11 individuals were determined to be purebred *D. suweonensis*, eight individuals were purebred *D. japonicus*, and 38 individuals were hybrids (Fig. 3B). Individuals with a Q value of 0.750 or higher for one species and a Q value of less than 0.250 for the other species in the STRUCTURE analysis were assumed to be backcrosses (*Weetman et al., 2014*). Therefore, 32 hybrid individuals were backcrossed using the above criteria. Among them, 13 individuals were assumed to be backcrossed with the maternal parentage by the mtDNA haplotype of *D. japonicus* and 19 of *D. suweonensis*, representing 56.14% of the total individuals.

For most individuals, the HRM analysis of the mitochondrial 12S rRNA gene and the ratio of Q values from the STRUCTURE analysis of the five nuclear genes were consistent with their maternal parentage determination. For example, most individuals with maternal inheritance of *D. japonicus* had a Q value of 0.721 or higher for the corresponding

species, and most individuals with *D. suweonensis* had a Q value of 0.716 or higher for the corresponding species. However, two individuals, SJ02_5 and JN02, were determined to be *D. japonicus* as maternal inheritance, but STRUCTURE analysis showed that their Q values of 0.406 and 0.346, respectively, were attributable to *D. japonicus*, indicating a least admixed individual or cyto-nuclear discordance.

## DISCUSSION

In general, nuclear DNA (nuDNA) is stably transmitted to the offspring and is characterized by biparental inheritance, whereas mtDNA is characterized by maternal inheritance. Methods that analyze both mtDNA and nuDNA for species and hybrid identification have been shown to significantly increase the accuracy in determining their parental lineage (*McKay & Zink, 2010*; *Sun & Pang, 2013*; *Toews & Brelsford, 2012*; *Whittaker, Assinder & Shaw, 1994*; *Funk & Omland, 2003*). Studies identifying hybrids through mtDNA-nuDNA comparative analysis have been used to identify introgressive hybrids to elucidate the process of introgressive hybridization and understand the level of genetic diversity (*Zhang et al., 2018*). These methods have been used in amphibian researches, including the identification of potential polyploid hybrids and backcrosses (*Correa et al., 2012*; *Stöck et al., 2010*; *Velo-Antón et al., 2021*). In this study, we used the mitochondrial 12S rRNA gene of *D. suweonensis*, *D. japonicus*, and their hybrids to identify maternal parentage by HRM analysis and applied DNA sequencing to five nuclear genes that contained SNPs, thereby greatly improving the identification accuracy of the purebred species and hybrids.

Sequence chromatograms from the five nuclear genes have the advantage of being able to reconstruct parental sequences of DNA segments from heterozygotes and interspecies hybrids for multiple linked points through the identification of SNPs and heterozygous sequence patterns (*Sousa-Santos et al., 2005*). While interspecific $F_1$ hybrid individuals are commonly characterized by heterozygous sequences at all SNPs between the two species (*Depaquit et al., 2019*; *Sousa-Santos et al., 2005*). However, the individuals analyzed in this study showed an irregular pattern. Most of the hybrids in this study were backcrosses.

*VÄHÄ & Primmer (2006)* employed two Bayesian-based programs, STRUCTURE and NEWHYBRIDS to effectively detect hybrids. They determined the optimal genetic differentiation threshold based on three key aspects, efficiency, accuracy, and overall performance, with a Q value of 0.900 or higher. In a previous study, hybrids of *D. suweonensis* were classified when the assignment probability was below 90.0% (*Borzée et al., 2020*). However, our study proposed a Q value threshold of 0.995 because of the absence of heterozygous sequences between *D. suweonensis* and *D. japonicus* across the five nuclear genes. Although reducing the threshold might result in a larger number of individuals being classified as purebred parental species, we argue that a stricter threshold aligns more closely with the criteria essential for determining parental species, particularly in the context of conserving endangered wildlife species (*De Hert et al., 2012*; *Yan et al., 2017*).

Admixture analysis can be used to identify $F_1$ and $F_2$ hybrids, and first-generation backcrosses, which are characterized by a decrease in the admixture rate (Q) of the species

by approximately one-half with each new backcross generation (*VÄHÄ & Primmer, 2006*; *Weetman et al., 2014*). In this study, we were able to accurately identify hybrids between *D. suweonensis* and *D. japonicus*, and the proportion of individuals that could be presumed to be backcrosses was very high, at 56.14% of the total individuals. In particular, the fact that no $F_1$ hybrids were identified between *D. suweonensis* and *D. japonicu* s suggests that hybridization between the two tree frog species has occurred over a long period of time, and hybrids can interbreed with one of their parental species that share the same habitats and that these species are not completely isolated and are largely admixed. Therefore, as mentioned above, it emphasizes the need to apply strict threshold values when separating the two tree frog species and their hybrids by Q values in STRUCTURE analysis, using the method developed in this study.

Hybridization, which is recognized as a significant driver of extinction, can imperil endangered wildlife species through processes such as hybridization suppression or genetic assimilation. The impact of hybridization on species can be detrimental, as it facilitates gene flow between different species, potentially leading to reduced biodiversity through direct and indirect pathways or even culminating in species extinction (*Levin, 2002*; *Rieseberg & Carney, 1998*). Our findings indicate that substantial hybridization or backcrosses between *D. suweonensis* and *D. japonicus* could markedly diminish their populations. Moreover, half of all individuals were backcrosses. Backcrosses tend to exhibit a higher imbalance in mitochondrial/nuclear ratios owing to paternal leakage, which detrimentally influences their survival (*Vilaça et al., 2023*). In the present study, two individuals of *D. japonicus* displayed a high level of admixture, with *D. Japonicus* mtDNA haplotype and a predominantly *D. suweonensis* nuDNA background. Similar instances of genetic mismatches between mtDNA and nuDNA have been observed in other amphibian species (*Ambu et al., 2023*; *Cairns et al., 2021*; *Eto, Matsui & Sugahara, 2013*). *Majtyka et al. (2022)* showed that the introgression of mtDNA was a consequence of repeated backcrossing and some hybrids between *Hyla arborea* and *H. orientalis* exhibited instances of cyto-nuclear discordance. The occurrence of cyto-nuclear discordance may suggest factors such as introgression through hybridization (*Lee-Yaw et al., 2019*), sex-biased dispersal (*Seixas, Boursot & Melo-Ferreira, 2018*), or shifts in hybrid zones (*Wielstra, 2019*). Notably, amphibians have demonstrated differential growth in individuals based on their mitochondrial type (mitotype), with lower growth rates occurring during instances of cyto-nuclear discordance (*Lee-Yaw, Jacobs & Irwin, 2014*). This is not only the case of cyto-nuclear discordance, but also the case of the presence of nuclear copies of mitochondrial pseudogenes (NUMTs) (*Hlaing et al., 2009*) that could be considered. Therefore, future studies should comprehensively clarify whether such genotypes are related to habitat or environmental factors.

## CONCLUSIONS

Previous studies have underscored that hybridization between *D. suweonensis* and *D. japonicus* constitutes a primary driver of the extinction threat faced by the former species (*Borzée, Andersen & Jang, 2018*; *Borzée et al., 2020*). Given the substantial number of hybrid individuals in this study, it is imperative to explore strategies to curb hybridization and safeguard endangered wildlife species. Hence, the crucial course of action involves elucidating the mechanisms underlying hybridization and promoting population stabilization (*Bohling, 2016*). The DNA sequencing strategy of the five nuclear genes used in this study is expected to offer several benefits. This can counteract the potential data bias attributed to null alleles when solely employing microsatellite markers for hybridization analysis. Additionally, nuclear gene markers can enable the stringent determination of purebred parental species with a higher resolution, thus significantly aiding in unraveling the mechanisms of hybridization. This study lays the groundwork for systematic investigations, enhancing the precise identification of purebred parental species and their hybrids, *D. suweonensis* and *D. japonicus*. Such advancements serve as a fundamental framework for guiding efforts toward the restoration of reproductive processes, a critical endeavor that necessitates the conservation and restoration of endangered *D. suweonensis*.

### Funding

This work was supported by a grant from the National Institute of Ecology (NIE), funded by the Ministry of Environment (MOE) of the Republic of Korea (NIE-B-2023-45). The funders had no role in study design, data collection and analysis, decision to publish, or preparation of the manuscript.

### Grant Disclosures

The following grant information was disclosed by the authors:
The National Institute of Ecology (NIE).
The Ministry of Environment (MOE) of the Republic of Korea: NIE-B-2023-45.

### Competing Interests

Keun-Yong Kim and Jung Soo Heo are employed by AquaGenTech Co., Ltd. The authors declare there are no competing interests.

### Author Contributions

- Nakyung Yoo performed the experiments, analyzed the data, prepared figures and/or tables, authored or reviewed drafts of the article, and approved the final draft.
- Ju-Duk Yoon conceived and designed the experiments, authored or reviewed drafts of the article, and approved the final draft.
- Jeongwoo Yoo analyzed the data, prepared figures and/or tables, and approved the final draft.

- Keun-Yong Kim performed the experiments, authored or reviewed drafts of the article, and approved the final draft.
- Jung Soo Heo analyzed the data, authored or reviewed drafts of the article, and approved the final draft.
- Keun-Sik Kim conceived and designed the experiments, performed the experiments, analyzed the data, authored or reviewed drafts of the article, and approved the final draft.

### Animal Ethics

The following information was supplied relating to ethical approvals (i.e., approving body and any reference numbers):

National Institute of Ecology approved this research (NIEIACUC-2020-012).

### Field Study Permissions

The following information was supplied relating to field study approvals (i.e., approving body and any reference numbers):

Han River Basin Environmental Office (No. 2020-24), Geum River Basin Environmental Office (No. 2020-24), Jeonbuk Regional Environmental Office (No. 2020-22), Won-ju Regional Environmental Office (No. 2020-24).

### DNA Deposition

The following information was supplied regarding the deposition of DNA sequences:

All nuDNA data are available at NCBI: OR474555–OR474832.

### Data Availability

The Q value from STRUCTURE analysis, results of mitochondrial DNA analysis using melting temperature from HRM analysis, species identification throug nuDNA-mtDNA comaprison about *D. suweonensis* and *D. japonicus*, and their hybrids are available in the Supplemental File.

### Supplemental Information

Supplemental information for this article can be found online at http://dx.doi.org/10.7717/peerj.16728#supplemental-information.

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
