# Peer review of "Development of molecular identification methods for Dryophytes suweonensis and D. japonicus, and their hybrids"

_PeerJ, doi:10.7717/peerj.16728_

## Round 0.1 · original submission · Minor Revisions

Three reviewers agree that this manuscript is scientifically sound and worthy of publication once minor issues, mostly of grammar and style but also some technical issues, are dealt with. I look forward to seeing a revised version.

**Language Note:** The Academic Editor has identified that the English language must be improved. PeerJ can provide language editing services - please contact us at copyediting@peerj.com for pricing (be sure to provide your manuscript number and title). Alternatively, you should make your own arrangements to improve the language quality and provide details in your response letter. – PeerJ Staff

·

Basic reporting

I’m grateful to read and review such an interesting and well-composed manuscript. I give some specific suggestions below (numbers refer to line numbers in the manuscript), and I have also made additional suggested edits to the English language using track changes in the manuscript itself.

3-4: I think “purebred parental species of” could be removed from the title for clarity and brevity – their purebred status is implied by the following “…, and their hybrids”.

3-4: The title could be made more descriptive by adding the word ‘molecular’ (i.e. “Development of molecular identification methods…”), but this is just a suggestion.

31: Consider changing from “…identify purebred parental species and their hybrids prior to conservation plans such as…” to “…identify both purebred and hybrid individuals prior to conservation plans such as…”. I think this is clearer, as identification an individual level seems to be what it being described, and this will not increase the limited abstract word count.

32-33: The current wording implies that purebred individuals also have abnormal ovaries and gonads. I suggest changing to: “Hybrids of the two species often have abnormal ovaries and gonads…”

28-33: In addition to the two comments above, I would propose restructuring this paragraph in the following suggested way, which will improve readability and reduce the word count: “Because hybridization can reduce biodiversity or cause extinction it is important to identify purebred parental species and their hybrids prior to conservation plans. The Suwon treefrog, Dryophytes suweonensis, is an endangered wildlife species in Korea that shares its habitat and often hybridizes with the common treefrog, D. japonicus. In particular, D. suweonensis and D. japonicus, and their hybrids often have abnormal ovaries and gonads, which are known to be a source of extinction threat.

39: Should this read: “…determine their parental species and hybrid status”?

50: Same comment as for line 31.

50-52: In addition to the comment above, I suggest breaking this paragraph up into two sentences, as paragraphs comprised on a long, single sentence can be difficult to read. I suggest: “Therefore, it was possible to unambiguously identify the parental species and their hybrids using the HMR analysis and DNA sequencing methods we applied in this study. This will provide fundamental information for D. suweonensis conservation and restoration research.”

78: The use of the word ‘animals’ here is ambiguous and could imply that 10% of all animal individuals could be hybrids, though I think you are referring to animal species. I suggest replacing the term ‘animals’ with ‘animal species’ if that is the case.

80: Since you have defined hybridization as involving different species earlier in the paragraph, I suggest the word ‘interspecific’ here is redundant and can be deleted.

84: Same comment as for lines 31 and 50, but given the structure of this sentence I think it can be altered to read: “Purebred and hybrid individuals have been identified using…”.

92: Should “animals” be changed to “species”, or are you referring to individuals?

109-110: Same comment as for lines 31 and 50.

109-111: Unless you have performed an analysis of the efficacy of such research with and without the use of you identification technique, I don’t think “proving to be valuable information prior…” is accurate to say. I suggest the following: “This integrated approach facilitated the unambiguous identification of the purebred parental species and their hybrids, thus providing valuable information for conservation and restoration research of D. suweonensis.

116-117: I’m not sure what is meant by “…captive and management wildlife…”. It would be grammatically correct if changed to “…capture and management of wildlife…, but I am not sure if that accurately conveys your intended message.

123-125: Might it be useful to specify how that sample size was divided among each locality? For example: “…including Suwon (x) and Pyeongtaek (x) cities in Gyeonggi-do, Chungju city in Chungcheongbuk-do (x), Asan city in Chungcheongnam-do (x)…”.

129: I do not think that oral swabbing of frogs should be described as non-invasive. I accept its use as a valid DNA sampling technique, but the term ‘non-invasively’ should be removed.

137-140: The term ‘treefrog species’ here is ambiguous and potentially misleading. Presumably you did not download available sequences for every species that can be described as a tree frog, which comprise a very large and paraphyletic group. It would be best to be specific here – was it just the two focal Dryophytes species, or perhaps all available hylids?

261-262: Same comment as for lines 31 and 50.

271: Same comment as for lines 31 and 50.

331-332: Same comment as for lines 31 and 50.

115-121: This information appears to be the same as or at least very overlapping with that in lines 358-363. Unless this is a requirement of the journal, I would suggest removing this information from the methods section and keeping it only in the Animal Ethics section towards the end.

549-551: I’m not sure what the convention is for reporting these data, but at first glance the format of Table 2 is a bit confusing without rereading the relevant text in the main manuscript. I recommend amending the table description to read “Sequence nucleotide polymorphism (SNP) sites (left) and parsimony informative sites (PI) (right), separated by a forward slash, in five nuclear genes of purebred Dryophytes suweonensis, D. japonicus, and their hybrids”.

Figure 2: I found the order lettering of the subfigures in Figure 2 (i.e. (a), (b), (c)) is very small and difficult to distinguish from the beginning of the sequence at first glance. This made the references to them in the figure description confusing. I also notice that some are slight cut off and others cover small sections of the first base in the sequence. I suggest making these slightly larger and/or bolder and moving them to the white space directly beneath where they currently are.

Figure 3: The small text on 3a makes the graph difficult to interpret and the difference in text size between 3a and 3b seems unnecessarily distracting. I think the figure would benefit from increasing all alphabetical text except the species/hybrid designations at the bottom of 3b to the same size as “mtDNA” in 3b. The Y axis label on 3b would need to take up two lines, but I think that would look fine. I also think the numeric text on 3a should be increased to match that of 3b. Finally, it was unclear to me a first what purpose the s/j tiles above in 3b serve aside from illustrating the nuclear discordance, then I realised they must be the mitochondrial results. I think that could be made clearer in the image description.

Experimental design

General (multiple locations in manuscript): Is there a particular reason you designed new primers for nuclear amplicons rather than using existing primers? I am aware that some of the existing options do not amplify equally well across all anuran species – did you want to develop primers that were optimised for amplification with the focal taxon, or perhaps for a more informative section of the gene than what existing primers amplify? If so, it may strengthen your manuscript to specify this.

Validity of the findings

No comments.

Additional comments

I congratulate the authors on a very well-written manuscript that will provide valuable data for informing research and conservation for a threatened species.

It would be interesting to note whether any non-molecular characteristics corresponded with purebred or hybrid status, for example intermediate morphological or acoustic profiles. This could perhaps provide a pre-molecular filter when selecting individuals for further, similar research or for applied conservation strategies. However, I understand that this is outside of the scope of the current study and do not expect such information to be included in this manuscript if the authors prefer not to.

I look forward to seeing this great work published!

Reviewer 2 ·

Basic reporting

In line 29, the English species name "Dryophytes japonicus" is written as "common treefrog," but the IUCN Red List for Dryophytes japonicus uses the name "Japanese Treefrog." However, "common treefrog" on the IUCN Red List website refers to an entire group of amphibians. It would be better to use the full English species name as provided by the authors of the IUCN Red List. This is not a critical issue but rather a suggestion.

Literature:
Change the order of references 436 and 439:
Move "Lee-Yaw JA, Jacobs CG, and Irwin DE. 2014. Individual performance in relation to cytonuclear discordance in a northern contact zone between long-toed salamander (Ambystoma macrodactylum) lineages. Molecular Ecology 23:4590-4602" to position 436.
Move "Lee-Yaw JA, Grassa CJ, Joly S, Andrew RL, and Rieseberg LH. 2019. An evaluation of alternative explanations for widespread cytonuclear discordance in annual sunflowers (Helianthus). New Phytologist 221:515-526" to position 439.
So, the corrected chronological order should be:
Lee-Yaw JA, Jacobs CG, and Irwin DE. 2014. Individual performance in relation to cytonuclear discordance in a northern contact zone between long-toed salamander (Ambystoma macrodactylum) lineages. Molecular Ecology 23:4590-4602.
Lee-Yaw JA, Grassa CJ, Joly S, Andrew RL, and Rieseberg LH. 2019. An evaluation of alternative explanations for widespread cytonuclear discordance in annual sunflowers (Helianthus). New Phytologist 221:515-526.

Figure:
On the page with Figure 2, the graph needs to be revised since there is no horizontal axis line at the intersection of species D. suweonensis and C-myc.
To improve readability, slightly increase the font size in Figures 3a and 3b.

Experimental design

no comment

Validity of the findings

no comment

Reviewer 3 ·

Basic reporting

First round review of:
Development of identification methods for purebred parental species of Dryophytes suweonensis and D. japonicus, and their hybrids by Yoo et al.

The authors sampled Dryophytes tree frogs in various sites of Korea and designed a genetic identification protocol to separate between the rare and endangered D. suweonensis, the widespread D. japonicus and their hybrid and backrosses. The protocol is based on mitochondrial haplotype identification using discriminant HRM melting temperatures and the sequencing of 5 nuclear introns giving access to a few dozens of discriminant SNPs used in STURCTURE analysis to assess the genetic composition of sampled individuals. Given the difficulty to identify these species and the threats that hybridization poses to D. suweonensis this protocol will be useful for distribution mapping and conservation planning and therefore deserves publication in PeerJ. Still, some parts of the manuscript are a bit confuse and needs to be clarified. I would therefore recommend minor reviews for this manuscript.

Experimental design

The experimental design is well justified for the adressed research question and fills a gap in conservation measure design.

Validity of the findings

The designed protocol is well described and usefull for conservation priotities and underlying data is available. Conclusions are well stated and justified.

Additional comments

L29-32 Reformulate

L38-39 “to determine their parental species and hybrids.” Reformulate or change in “to determine their ancestry proportion or their hybrid status”

L43 “DNA sequencing the five” should be “DNA sequencing of the five”

L45 Reformulate: “inferred from the variant sites showed a delta K of two.” Should be “showed a two-group structure as the most likely grouping solution” or something approaching as the STRUCTURE programme doesn’t give DeltaK as output (calculated in STRUCTURE HARVESTER)

L45 “double peak” should be explained and/or replaced by “heterozygous position” which is much clearer.

L58-59 Could be worth to detail the threats.

L67-81 For reading clarity I would recommend to put this paragraph at first and then the one about the study species (L58-66) after.

L68 Hybridization is common and widespread in the wild (as written L76-77), human activities are not the main cause of it, should therefore be mitigated.

L82 Replacing “has been identified” by “has been conducted” for more clarity.

L91-93 These mtDNA characteristic are also problematic as they can result in flawed results, also mtDNA does not recombine. This sentence should be rephrased and mitigated.

L104-105 “to determine their parentage and hybridization.” Should be re-phrased, maybe as “to determine their parentage and genetic compositions”.

L107 I would replace “conservation researches” by “conservation measures”.

L112 “And We had approval for”: the word “And” should be deleted.

L157-173 You should put this “HRM” paragraph before the paragraph de-scribing nuDNA sequencing to cope with abstract and result order.

L203-204 Even if it’s well known you should explain what “double peak” mean regarding the DNA sequence (heterozygous position) and what it means regarding the individuals’ hybrid status.

L216-224 This paragraph is very complicated to follow and needs to be clar-ified. As you finally use Q > 0.995 to confirm that an individual is a pure-bred individual you should mostly discuss about that and less about Q values > 0.8 or 0.9 (just write that 16 individuals showed 0.8 or less, 23 showed 0.9 or less etc.).

L226-229 Clarify that it appears that most (all?) of the hybrids are back-crosses because ancestries differ from typical F1 50-50 ancestries. Explain how you assume that individuals are backcrosses with the maternal parent-age of one or the other of the species based on your nuDNA dataset. If it’s only using their mtDNA haplotype, please clarify it.

L235-238 Q = 0.4 or 0.3 is typical of backcrosses or at least admixted indi-viduals so no real cyto-nucl discordance just a mtDNA haplotype differing from the dominant genomic composition, please mitigate.

L242-243 mtDNA is useless for hybrid identification due to maternal trans-mission, please clarify the sentence.

L251-253 “HRN” should be “HRM”.
“and also applied the DNA sequencing of the five nuclear genes that con-tains SNP sites, thereby greatly improving the identification accuracy of the purebred parental species and hybrids.” Should be rephrased maybe as “and applied DNA sequencing to five nuclear genes which contain species-specific SNPs, thereby ….”.

L259-262 This sentence is very difficult to understand and needs to be clari-fied. It seems that most of the hybrids of this study are not F1 but more probably backcrosses therefore the fact that these individuals are not hetero-zygous at all species-specific SNPs seems normal.

L263 “Vähä& Primmer”: a space is lacking. + same at L276.

L268-270 Unclear sentence, explain the link between Q > 0.995 and the ab-sence of double peak.

L278 “reverse hybrid” should be replaced by “backcrosses”.

L280-281 And also that these species are not completely isolated and largely admix.

L291-292 You state above that all the 32 hybrids of the study are not F1 hy-brids and therefore are backcrosses, it’s therefore unclear why you write about 81.58% of backcrosses in this part. Please clarify/correct.

L292-293 Explicit why, incompatibilities between nuDNA backround and mtDNA? Breakdown of coadaptations?

L294 Can’t see anything above about these two individuals except if they are the two “cyto-nuclear discordance” cases discussed above. Exhibiting D. ja-ponicus mtDNA haplotypes with a mostly (approx. 0.6-0.7) D. suweonensis nuDNA background just makes these individuals highly admixed individuals as said above.

L297-298 “Introgression through hybridization” seems more correct. Could also be ILS (Nabohlz 2023, Biol J Linn Soc) or NUMTs even if unlikely here.

L306-309 “between D. suweonensis and D. japonicus that constitutes” delete the word “that”.

Table 1. Can’t find to what “* M = A + C, R = A” refers and please explain what “tm(°C) means.

Figure 2. Low quality makes difficult to read, can't see any letters “(a-c)”; “(d-f)”; etc. corresponding to the legend. The position of the species names is ambiguous because of a to small spacing between the parts of the figure.

Figure 3. “(a) The best suitable” maybe better as “(a) The most likely num-ber K of groups for D. suweonensis …”. Can’t properly see the “Red box” due to figure quality but as written above these highly admixed individuals doesn’t really represent cyto-nucl discordances.

---

## Round 0.2 · accepted · Accept

The authors have addressed all the reviewers' comments. The manuscript is greatly approved and now is suitable for publication.